# Self-assembly of nanocrystal checkerboard patterns via non-specific interactions

Yufei Wang[1,2,5], Yilong Zhou [3,5], Quanpeng Yang [3], Rourav Basak [4], Yu Xie[1], Dong Le[2,4], Alexander D. Fuqua[1], Wade Shipley[1,2], Zachary Yam[1], Alex Frano [4], Gaurav Arya [3] ✉ & Andrea R. Tao[1,2] ✉

Checkerboard lattices—where the resulting structure is open, porous, and highly symmetric—are difficult to create by self-assembly. Synthetic systems that adopt such structures typically rely on shape complementarity and site-specific chemical interactions that are only available to biomolecular systems (e.g., protein, DNA). Here we show the assembly of checkerboard lattices from colloidal nanocrystals that harness the effects of multiple, coupled physical forces at disparate length scales (interfacial, interparticle, and intermolecular) and that do not rely on chemical binding. Colloidal Ag nanocubes were bi-functionalized with mixtures of hydrophilic and hydrophobic surface ligands and subsequently assembled at an air–water interface. Using feedback between molecular dynamics simulations and interfacial assembly experiments, we achieve a periodic checkerboard mesostructure that represents a tiny fraction of the phase space associated with the polymer-grafted nanocrystals used in these experiments. In a broader context, this work expands our knowledge of non-specific nanocrystal interactions and presents a computation-guided strategy for designing self-assembling materials.

Self-assembly—where components spontaneously organize themselves—can be carried out on a massively parallel scale to construct large-scale architectures using colloidal nanocrystal building blocks. These colloidal nanocrystal systems minimally consist of a hard particle, particle-grafted ligands, and a carrier solvent, and have been demonstrated to self-assemble into beautifully ordered superlattices through evaporation of the solvent or via chemical binding between the grafted ligands[1–3]. These nanocrystals often exhibit complex emergent phase behavior due to their inherently non-rigid form and non-additive nanoscale interactions[4–7]. As a result, the valence and coordination geometry of nanocrystals within the resulting superstructure remain challenging to control and predict, even within relatively simple two-dimensional (2D) lattices.

Many synthetic systems have sought to harness the chemical principles inspired by biological self-assembly to generate porous 2D lattices, such as checkerboard-type structures. Zhang et al. demonstrated the assembly of a synthetic 2D protein array using patchy protein building blocks[8]. They modified a C4-symmetric protein with cysteine residues to dictate corner–corner binding with a valence of four. Lateral protein binding resulted in solution-grown 2D crystals that possess open square pores. Similar 2D arrays have been assembled using artificial DNA nanostructures as building blocks. For instance, Liu et al. used a double-layer cross-shaped DNA origami tile programmed with complementary single-stranded DNA sticky ends to generate a porous 2D DNA lattice[9]; integrating Au nanoparticles into the DNA origami tiles enabled the assembly of planar nanoparticle arrays including 2D square lattices[10]. The checkerboard-type structures generated in all these self-assembly systems required precise control of building block valence, which is facilitated by the following two design principles: (i) shape complementarity (i.e., square tiling)

[1]Department of Chemical and Nanoengineering, University of California San Diego, La Jolla, CA, USA. [2]Materials Science and Engineering Program, University of California San Diego, La Jolla, CA, USA. [3]Department of Mechanical Engineering and Materials Science, Duke University, Durham, NC, USA. [4]Department of Physics, University of California San Diego, La Jolla, CA, USA. [5]These authors contributed equally: Yufei Wang, Yilong Zhou. ✉e-mail: gaurav.arya@duke.edu; atao@ucsd.edu

and (ii) site-selective binding (i.e., corner–corner). Enacting these design principles has required the highly specific chemical interactions only available to biomolecular systems.

Similar porous, periodic, and interconnected structures have yet to be achieved with inorganic nanocrystals, which can offer a range of interesting material functions depending on nanocrystal composition and morphology, but do not intrinsically display the specific binding interactions characteristic of protein and DNA building blocks. Biomolecular binding interactions usually lead to assemblies that are fragile, placing severe limitations on solvent conditions and often requiring specific charge conditions to avoid denaturation (in proteins) and surface binding (in DNA tiles). Inorganic nanocrystal assembly requires more robust interparticle interactions that can accommodate a wider swath of experimental conditions. Typical self-assembly approaches to controlling nanocrystal valence use either anisotropic particle shapes (e.g., nanocubes and nanorods) or anisotropic surface chemistries (e.g., patchy colloids[11]). However, even these highly anisotropic nanocubes (NCs) are subject to strong driving forces for crystallization that produce dense, close-packed assemblies rather than optimally interconnected nanocrystals. The ability to generate precisely interconnected nanocrystal lattices using solely non-specific chemical interactions—which are prevalent and compatible with any inorganic nanocrystals—would therefore be highly desirable.

Here we report the self-assembly of inorganic checkerboard lattices by harnessing the competition across several types of non-specific intermolecular interactions associated with polymer-grafted metal nanocrystals. Interfacial forces, entropy-driven steric forces, hydrophobic forces, and particle shape are all chemically programmed and integrated to rationally design nanocrystal binding, valence, and orientation (Fig. 1a). To generate large-scale 2D arrays, we carried out nanocrystal self-assembly at an air–water interface, which bypasses substrate interactions or solvent restrictions associated with evaporation-induced assembly[12,13]. Colloidal Ag NCs are used as the core nanocrystal building block because of their anisotropic shape, which allows them to adopt distinct orientations with respect to the interface (e.g., face-up, edge-up, or vertex-up)[14,15] and mediate different interparticle connections (e.g., face–face or edge–edge)[16]. We modified the surface of the Ag NCs using a mixture of two ligands: a short predominantly hydrophobic graft, which introduces attractive interactions between NCs, and a long predominantly hydrophilic graft, which introduces steric repulsion between NCs. Having two species of grafts allows us to simultaneously control particle orientation and NC connectivity, which is challenging to achieve with a single-graft system.

## Results

### Computation-guided assembly of bi-grafted Ag NCs

Figure 1b shows scanning electron microscopy (SEM) images of a checkerboard lattice obtained using our materials design approach, where iterative feedback between coarse-grained molecular dynamics (CG MD) simulations and nanocrystal assembly experiments was used to converge upon the appropriate assembly conditions (see details in Supplementary Note 4). Ag NCs were modified using a ligand feedstock consisting of a mixture of thiolated polyethylene glycol chains (PEG-SH) and 1-hexadecanethiol ($C_{16}$-SH). The resulting mesoscopic assemblies (here deposited onto solid support) exhibit edge–edge NC connections with a valence of four and an almost perfect 90° internal bond angle. Figure 1c, d shows the corresponding structure predicted by CG MD simulations, which further corroborates that the NCs are oriented face-up at the air–water interface. Figure 1e shows the 2D Fourier transform of an SEM image of the checkerboard lattice, from which we extract the local four-fold NC symmetry using angular cross-correlation (ACC) analysis[17–19] (Fig. 1f).

The checkerboard lattice represents only a small fraction of the rich structural phase space that can be accessed using our NC

assembly system. Figure 2a shows the CG model used for building the mesostructural assembly phase diagram, where Ag NC cores were modeled as rigid lattices, the hydrophobic and hydrophilic ligands as flexible bead chains of lengths $l_{Ho} = 3\sigma$ and $l_{Hi} = 6\sigma$, and the solvent molecules as single beads of size $\sigma$, which sets the length scale of the system (Supplementary Fig. 3). Appropriate interbead potentials were implemented to capture van der Waals attraction between the NC cores, attractive interactions between hydrophobic ligands, steric repulsion of the hydrophilic ligands, and surface tension of the air–water interface (see "Methods" and Supplementary Note 7 for details). Figure 2b shows the phase diagram plotted as a function of two system parameters: (i) the overall graft density of the ligands $\Gamma_g$ (in units of CG bead chains per $\sigma^2$) and (ii) the number percentage of hydrophobic ligands on the NC surface. At each condition, MD simulations were carried out using the CG model for 16 grafted NCs (Supplementary Fig. 4), and the resulting structures were qualitatively categorized as dispersed, 1D or 2D, then further distinguished by NC connectivity and interfacial orientation to yield a total of six distinct phases (i.e., face–face 2D, checkerboard, face–face 1D, edge–edge 2D, edge–edge 1D, and dispersed).

The checkerboard lattice only forms when the NCs are grafted with a dense ligand corona of high hydrophobic content. These conditions lead to a face-up orientation of isolated NCs (Fig. 2c), a key prerequisite for checkerboard formation; the CG model shows that these face-up NCs are submerged ≈80% into the water subphase (Fig. 2c, inset), as confirmed by optical spectroscopy (Supplementary Fig. 5). Even though the two ligands are uniformly distributed on the NC surface, the longer hydrophilic ligands stretch away from the NC surface to minimize their interaction with the shorter hydrophobic ligands and maximize their interaction with water. The stretching of the hydrophilic ligands, together with their low graft density, exposes the NC edges which are primarily hydrophobic (Fig. 2c inset and Supplementary Fig. 6). Figure 2d shows the orientational free energy landscape computed for a single NC as a function of Euler angles ($\theta,\varphi$) (Fig. 2c inset), affirming that the face-up orientation is indeed the most stable compared to the edge- and vertex-up orientations. Analysis of ligand density distribution (Fig. 2d insets) reveals that when considering both the cross-sectional area of the NC core and the stretching of the hydrophilic ligands at the interfacial plane, the face-up orientation maximizes the occluded interfacial area. The attraction between NCs via their exposed hydrophobic edges combined with the steric repulsion from hydrophilic ligands (which prevents face–face contact) is what facilitates the formation of ≈90° edge–edge contacts and an NC binding valence of four. Free energy calculations for two approaching NCs (Fig. 2e), along with decomposing the total free energy into contributions from the hydrophobic and hydrophilic ligands, demonstrate that edge–edge contacts—where steric repulsion from hydrophilic ligands is weak—are favored. For face–face contacts, steric repulsion overwhelms the attractive contribution from the hydrophobic ligands. Assembly into a checkerboard lattice requires a delicate balance of these two interactions: traversing the phase diagram to a lower hydrophobic content reduces attraction between NCs, leading to dispersed or 1D morphologies; traversing the phase diagram to higher hydrophobic content and/or lower $\Gamma_g$ leads to face–face connections and more compact morphologies (for more details, see Supplementary Note 7 and Supplementary Fig. 8).

### Experimental validation of the NC mesophase diagram

To experimentally access this phase space, we generated a combinatorial library of polymer-grafted Ag NCs of different NC core sizes and ligand lengths, chemistries, and graft densities. We synthesized colloidal Ag NCs (length ≈ 80 nm) and carried out ligand exchange reactions to displace NC capping agents with a mixture of hydrophilic (PEG) and hydrophobic (alkyl chains or polystyrene) ligands. First, we evaluated assembly structures obtained for Ag NCs incubated with a

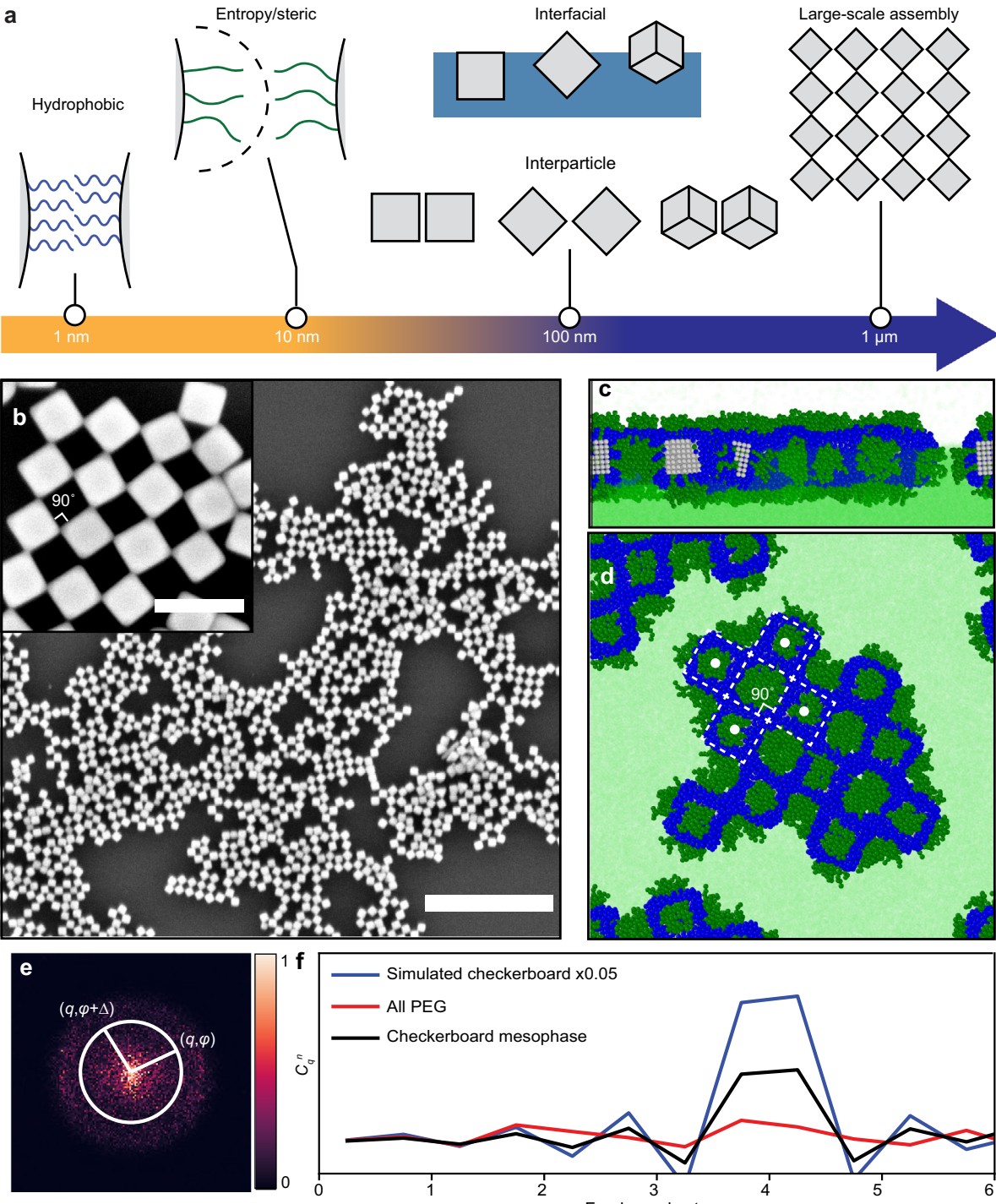

**Fig. 1 | Self-assembly of checkerboard mesophase from Ag nanocrystals.**
**a** Design strategies operating at different scales were harnessed together to achieve the assembly. **b** SEM image of the mesophase constructed with edge–edge connected checkerboard lattice obtained with Ag NCs post-synthetically modified with a feedstock mixture of 50 µM PEG20k-SH and 6 µM $C_{16}$-SH. Scale bar = 500 nm, inset = 100 nm. **c, d** Simulation snapshots of the assembled NC checkerboard lattice at an interface with side (**c**) and top (**d**) views. The PEG20k-SH, $C_{16}$-SH, and water are represented by dark green, blue, and light green, respectively. **e** A 2D Fourier transform magnitude of an SEM image over a larger field of view (4.5 µm x 4.5 µm) was used to quantify the extent of local order by evaluating the angular cross-correlation (ACC) function $C_q(\Delta)$ between the intensity at two points

separated by an angular separation, $\Delta$, at a given $q$ value inversely proportional to the spatial length scale of the order being probed. The details of this analysis are provided in Supplementary Figs. 1 and 2. **f** A second Fourier transform is taken on the ACC function $C_q(\Delta)$ to extract the symmetry order n of the function with respect to $\Delta$, $C_q^n$ (Supplementary Note 3). An increase in the four-fold symmetry term $n = 4$ in samples with PEG:$C_{16}$ = 50 µM:6 µM ligand feedstock (green) indicates the presence of checkerboard order compared to the all-PEG sample (red). For reference, the blue line shows $C_q^n$ evaluated for a simulated perfect checkerboard pattern, scaled down by 0.05 for easy comparison. Source data are provided as a Source Data file.

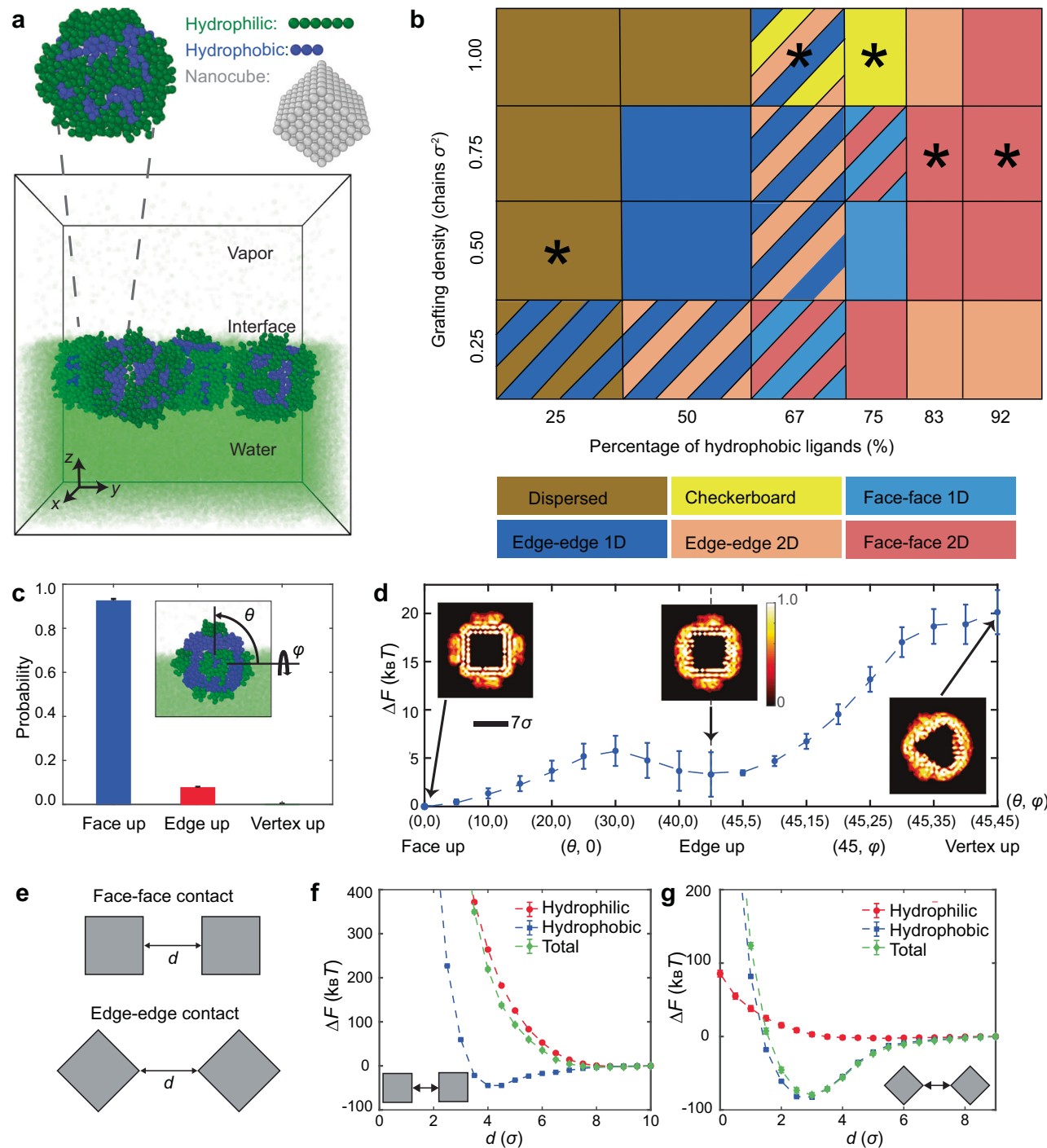

**Fig. 2 | Predicted assembly behavior of NCs with two species of grafts at an interface. a** Coarse-grained model of air–water interface and bi-grafted NCs, where the water layer is shown in fluorescent green, the PEG-SH ligands are shown in green, the $C_{16}$-SH ligands are shown in blue, and the NC cores are shown as cubic lattices of gray beads. **b** Structural phase diagram as a function of overall graft density $\Gamma_g$ and number percentage of $C_{16}$-SH ligands $C_{16}/(C_{16} + PEG)$. The phases labeled by asterisks are experimentally achieved through modifying ligand concentration in feedstock and are shown in Fig. 3a–e. **c** Probability of NCs exhibiting the three principal orientations obtained from a freely mobile NC with $\Gamma_g = 1.0$ chains/$\sigma^2$ and $C_{16}/(C_{16} + PEG) = 75\%$ at the interface. Inset: a representative snapshot of the NC at the interface captured from the simulation. The corresponding probability with $\Gamma_g = 1.0$ and $C_{16}/(C_{16} + PEG) = 25\%$ is provided in Supplementary Fig. 7. **d** Free energy profiles $\Delta F$ of the NC with its center constrained at equilibrium vertical position as a function of two rotation angles $(\theta, \varphi)$, as defined in (**c**). Insets show the graft segment density around NCs at the interfacial plane for the three principal orientations, showing their occluded interfacial areas. **e** Free energy profiles $\Delta F$ of two face-up oriented NCs constrained at their equilibrium vertical position as a function of surface–surface distance $d$ for face–face (**f**) and edge–edge contacts (**g**). Error bars in (**d**, **f**, **g**) are derived from the standard error, based on 50,000 independent samples. Source data are provided as a Source Data file.

ligand feedstock (in molar excess) prepared at different molar ratios of PEG-SH ($M_w = 20,000$ g mol⁻¹) and 1-hexadecanethiol ($C_{16}$) to determine the experimental conditions that best achieve the predicted checkerboard phase. Total graft density measured by inductively coupled plasma mass spectrometry (ICP-MS) ranges from 0.826 to 1.636 ligands nm⁻², consistent with values expected for NCs functionalized with these two disparate ligands (Supplementary Table 1). Hydrodynamic radii of the different PEG-grafted NCs measured by

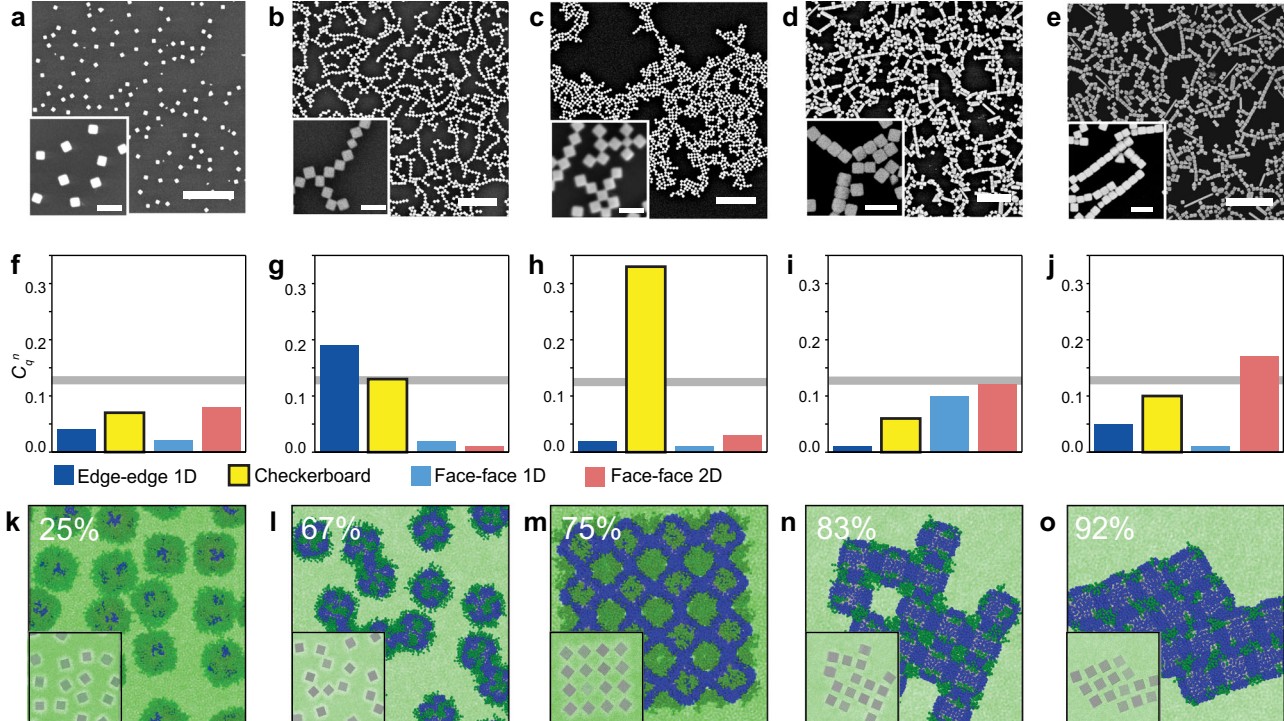

**Fig. 3 | Experimental validation of the Ag NC mesophase diagram. a–e** SEM images of the assembly results using Ag NCs post-synthetically modified with the following feedstock concentrations: **a** 0 μM, **b** 3 μM, **c** 6 μM, **d** 9 μM, and **e** 15 μM $C_{16}$ and 50 μM PEG20K. Scale bar = 1 μm, inset = 200 nm. **f–j** Histogram of the Fourier coefficients $C_q^n$ (unitless) of phases obtained from ACC analysis of each SEM image (**a–e**), with the threshold value for determining the dominant mesophase marked by the grey line. **k–o** Simulation results of assemblies composed of 16 Ag NCs with overall graft density $\Gamma_g = 0.75$ or 1 and varying $C_{16}$/($C_{16}$ + PEG) content (percent coverage shown) corresponding to each SEM image (**a–e**). The insets show NCs without grafts for better visualization. Source data are provided as a Source Data file.

dynamic light scattering show that the grafted NCs possess a PEG–core size ratio in the range of 0.255–0.375 (Supplementary Table 2), closely matching the size ratio in CG MD simulations. Assembly was carried out by dispersing the functionalized NCs onto an air–water interface and allowing the NCs to self-organize over a period of hours to days. To characterize interfacial assembly, we carried out electrodynamic modeling and measured the UV-Vis reflection of NCs at the air–water interface (Supplementary Fig. 5). The resulting NC structures were transferred to a Si substrate via Langmuir–Blodgett deposition and imaged by SEM. ACC analysis was performed to identify specific mesophases with values of $q$ corresponding to the edge–edge $1/(2a)$ or face–face $1/(3a)$ connections and values of $n$ corresponding to 1D ($n = 2$) and 2D ($n = 4$) symmetries (see Supplementary Note 3 for detailed methods).

Figure 3a–e shows SEM images of NC assemblies obtained for NCs functionalized with PEG20K (50 μM) at a fixed concentration and increasing $C_{16}$ concentration in the ligand feedstock, corresponding to the following mesophases: (a) dispersed (0 μM, $\Gamma_g = 0.5$ nm$^{-2}$), (b) edge–edge 1D (3 μM, $\Gamma_g = 1.6$ nm$^{-2}$), (c) checkerboard (6 μM, $\Gamma_g = 1.4$ nm$^{-2}$), (d) face–face 2D (9 μM, $\Gamma_g = 1.1$ nm$^{-2}$), and (e) face–face 2D (15 μM, $\Gamma_g = 0.8$ nm$^{-2}$). Figure 3f–j shows the histogram of the Fourier coefficients obtained for each NC assembly. Based on $\Gamma_g$ and this image analysis, we determine that this NC library traverses the phase diagram roughly along the starred mesophases shown in Fig. 2b. Representative NC morphologies obtained from CG MD simulations corresponding to the points in Fig. 2b are shown for comparison (Fig. 3k–o). Assembly experiments using NCs composed of different NC sizes (60–100 nm, Supplementary Fig. 9), and different hydrophobic ligand chemistries (1-octadecanethiol, polystyrene thiol, and 2-naphthalenethiol, Supplementary Fig. 10) show that NCs adopt similar mesophases, *ceteris paribus*. However, checkerboard NCs are only observed for a fraction of samples obtained under similar feedstock mole ratios as the NCs in Fig. 3a–e, with edge–edge 1D NCs observed as the predominant mesophase. This confirms the narrow window of experimental NC parameters for checkerboard lattice formation, consistent with the phase diagram in Fig. 2b.

**Defects and long-range order of the checkerboard mesophase**
Low magnification SEM images show that the checkerboard lattice formation is not limited to a small region of the sample, and rather, represents the major product of assembly (Fig. 4a). Although the domain sizes of the checkerboard motifs observed in the experiments were small, such assemblies can be captured from virtually everywhere on the macroscopic air–water interface (Supplementary Fig. 11). We did observe, however, that many of our experiments led to edge–edge 1D and face–face 1D mesophases in coexistence with the checkerboard motifs. This is also consistent with our phase diagram, which shows a transition band where multiple assembly morphologies coexist. Most importantly, we found that this coexistence is prominent for NCs grafted with a 7:3 hydrophobic to hydrophilic ligand ratio and ≈0.75 normalized graft density (Fig. 4b), confirming the sensitive nature of checkerboard lattice formation. While atomic force microscope characterization (Supplementary Fig. 12) confirms that PEG chains are well distributed on the Ag NC surface, small variations in graft density or graft distribution (Supplementary Fig. 13) increase the likelihood that 1D assemblies are formed.

Checkerboard lattice formation is also hindered by two types of assembly defects shown in Fig. 4c: vacancies (where an NC is missing from the lattice) and triangles (where three NCs assemble via edge–edge connections). Our CG MD simulations reveal that checkerboard assembly proceeds through a cluster–cluster aggregation

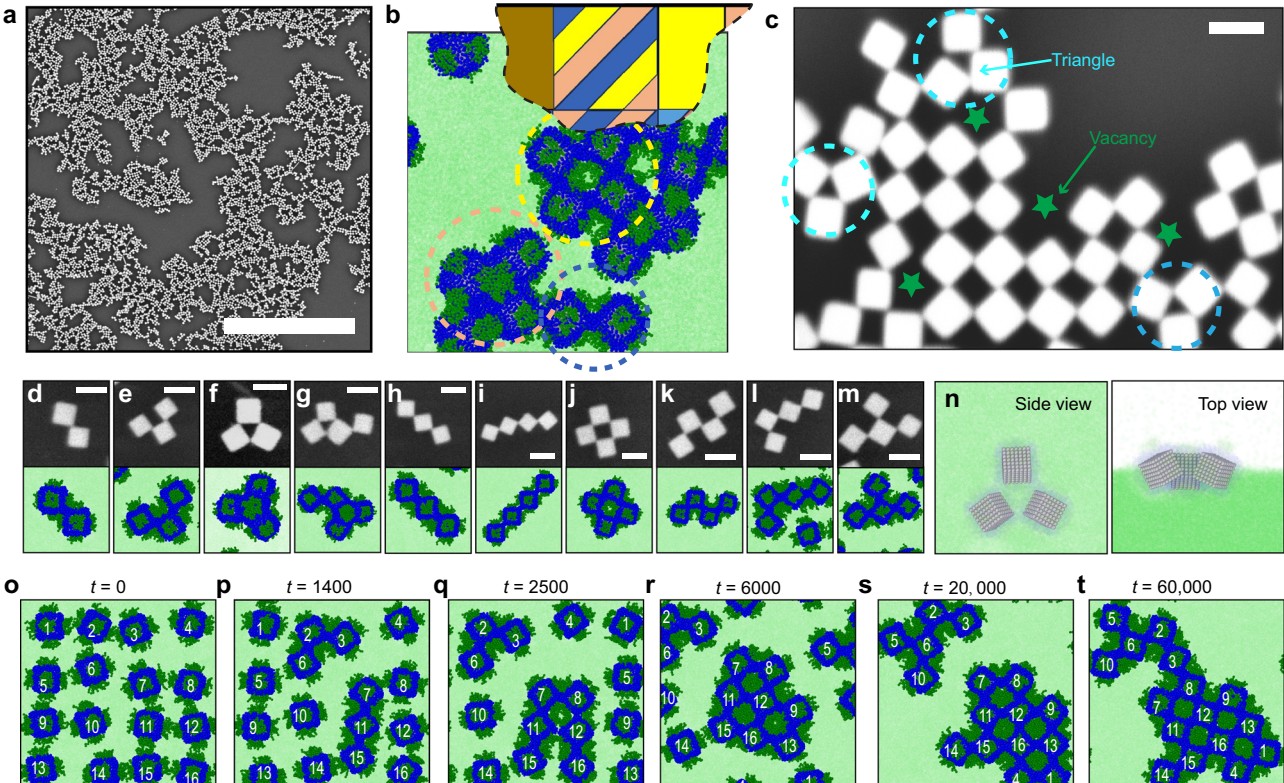

**Fig. 4 | Defects in the assembly of checkerboard lattices and corresponding optimization. a** Low magnification SEM image of checkerboard mesophase. Scale bar = 10 μm. **b** Part of the self-assembly phase diagram in Fig. 2b was constructed with 16 NCs and an example assembly pattern consisting of three different phases and two defects in the condition of $\Gamma_g = 1.0$ chains/$\sigma^{-2}$ and the number percentage of hydrophobic ligands = 67% in the simulation. The circles with certain colors are AgNC assemblies that correspond to the same color phase in the phase diagram. **c** Two types of defects commonly realized in Ag NC assembly: vacancies (green) and triangle (cyan) arrangements. **d**–**m** Catalog of NC clusters (≤4 NCs) that have been realized in both experiments and simulations. Scale bar = 100 nm. **n** Top and side view of the triangle defect composed of three edge–edge connected NCs at the air–water interface. **o**–**t** Assembly pathway for 16 grafted NCs with $l_{Hi} = 8\sigma$ and with respect to simulation time $t(m\sigma^2/\varepsilon)^{1/2}$. NCs are labeled by numbers.

mechanism and that, much like a poor game of Tetris, vacancies emerge when NC clusters that are not shape complementary assemble. Figure 4d–m presents a catalog of NC clusters (≤ 4 NCs) observed in both experiments and CG MD simulations. Calculations confirm that the checkerboard cluster (Fig. 4j) exhibits the lowest free energy ($-108 \pm 11\,k_B T$) amongst similarly sized clusters, which represent metastable states (Supplementary Fig. 14). Except for the triangle-shaped clusters (Fig. 4f, g), all of the NC clusters are potential "seed" structures for a checkerboard lattice, which can form if NC clusters connect to each other with complementary shapes or if vacant sites in the NC clusters are filled by freely diffusing, individual NCs. Triangle defects have a similarly large but slightly less favorable free energy ($-97 \pm 7\,k_B T$) than the checkerboard cluster. Thus, they are irreparable during the assembly process. Once formed, they disturb the structural arrangement of nearby NC clusters (Fig. 4c and Supplementary Fig. 14). A closer investigation of this defect reveals that the three NCs are connected through edge–edge contacts that possess a compressed internal angle and puckered orientation at the air–water interface (Fig. 4n and Supplementary Fig. 15). Even though this requires the NCs to rotate slightly away from the most-favored face-up orientation (Fig. 2d), this rotation alleviates some of the steric repulsion between connected NCs.

As such, we investigated if triangle defects could be prevented by increasing the hydrophilic ligand length to maximize the angle of contact between connected NCs and increase the energy cost of NC rotation. Increasing hydrophilic graft length from $l_{Hi} = 6\sigma$ to $8\sigma$ causes all NCs to now participate in checkerboard lattice formation without any triangular defects (Supplementary Fig. 16). Figure 4o–t

shows snapshots of the assembly process for 16 NCs grafted with these longer hydrophilic ligands. The NCs assemble into small clusters of various shapes (e.g., the flat-topped ridge formed by NCs 2–3–6 and the linear string formed by NCs 7–11–15) that eventually grow and merge into a checkerboard lattice. At $l_{Hi} = 9\sigma$, the large steric repulsion between NCs becomes too large, causing the NCs to assemble into an edge–edge 1D mesophase (Supplementary Fig. 16d). This defect suppression through tuning $l_{Hi}$ is confirmed by assembly experiments using NCs with PEG grafts of varying $M_w$ (10,000–54,000 g mol⁻¹, Supplementary Figs. 17 and 18).

## Discussion

Our combined experimental and simulation results demonstrate that non-specific chemical interactions can be used to generate bespoke, precisely interconnected nanocrystals. The checkerboard-type structures achieved here have two key functional features: (i) shaped pores that have the potential to be used in chemical separations and/or capture, and (ii) strong coupling of quadrupolar plasmonic resonances that give rise to unique optical behavior (Supplementary Fig. 5). Control over coupled interfacial and interparticle, and intermolecular forces was key to our approach and for achieving the self-assembly of low- and intermediate NC density mesophases. We used CG models as an efficient tool for rapidly exploring this phase space and identifying the key experimental parameters of nanocrystal building blocks (chemical composition, nanoscale dimensions, and surface graft densities) for assembling a periodic structure. The assembly of inorganic NCs into checkerboard lattices—a small and singular target in a vast mesophase space—shows the potential for this bullseye approach in

the design of synthetic mesoscopic architectures. The two-graft strategy used in this study is quite simple but effective at providing a knob for tuning coupled interfacial–interparticle interactions. These surface chemistries should be highly generalizable to organizing other inorganic matter into similar architectures for applications in nanomanufacturing[20], sensing[21,22], catalysis[23,24], and photovoltaics[25].

# Methods

## Chemicals
Silver nitrate (≥99%), 1,5-pentanediol (PD, ≥97%), copper(II) chloride (≥98%), poly(vinylpyrrolidone) (PVP, average $M_w$ = 55,000 g mol$^{-1}$), poly-(ethylene glycol) methyl ether thiol (PEG, average $M_w$ = 6000 g mol$^{-1}$), 1-hexadecanethiol ($C_{16}$), 1-octadecanethiol ($C_{18}$), and 2-naphthalenethiol were purchased from Sigma-Aldrich. Poly-(ethylene glycol) methyl ether thiol (PEG, average $M_w$ = 10,000–30,000 g mol$^{-1}$) were purchased from Laysan Bio. Poly-(ethylene glycol) methyl ether thiol (PEG, average $M_w$ = 54,000 g mol$^{-1}$) and ω-thiol-terminated poly(styrene), were purchased from Polymer Source. Water used in experiments was obtained from a Millipore water purification system with a resistivity of 18.2 MΩ cm.

## Synthesis and purification of Ag NCs
Ag NCs were synthesized using a previously reported polyol reaction[26]. The AgNO$_3$ precursor solution was prepared by adding 40 μL of 0.043 M CuCl$_2$ solution into 0.20 g AgNO$_3$ dissolved in 5 mL of 1,5-pentanediol and sonicating until all the salt crystals were dissolved. 0.20 g PVP was dissolved in 10 mL pentanediol. The reaction solution was prepared by heating 10 mL pentanediol in a 50 mL glass round bottom flask under continuous stirring in an oil bath heated to 195 °C. The AgNO$_3$ and PVP solutions were alternately injected into the RBF at a rate of 500 μL min$^{-1}$ and 320 μL per 30 s five times. The heating was then stopped, and the dispersion was cooled down to room temperature. The dispersion was then filtered using membranes with pore sizes 650 nm, 450 nm, and 220 nm to remove nanowires and large nanoparticles. The filtered dispersion was centrifuged and redispersed at an optical density of 60 abs. u. in chloroform with a small amount of ethanol added to stabilize.

## Surface modification of Ag NCs
During the ligand exchange reaction with two ligand mixtures, the number of PEG-SH ligands added in the feedstock was kept constant at 50 μM and the amount of hydrophobic ligand was tuned from 1 μM to 15 μM. It is worth noting that the solvent used in the ligand exchange with pure PEG-SH was water, because PEGylation of the Ag NCs did not happen when using chloroform as solvent. Ligand exchange took place for 72 h and was followed by centrifugation to remove excess ligands. The sediment was redispersed in chloroform for further assembly experiments and in water for DLS and ICP-MS measurements.

## Interfacial self-assembly
To create a large area of the 2D monolayer, functionalized Ag NCs in chloroform were dropped casted onto the surface of the water in a glass petri dish. After the chloroform evaporated, the petri dish was covered, and the monolayer was allowed to assemble for days. The assembled monolayer was transferred onto the surface of the silicon wafer through dip coating vertically and imaged by SEM.

## Characterization
SEM characterization was carried out using FEI Apreo SEM at an accelerating voltage of 5 kV. Hydrodynamic diameter values were measured using a Malvern NANO-ZS90 Zetasizer. Ligand packing density was analyzed by ICP-MS.

## Coarse-grained model
To investigate the dynamics of NCs grafted with two species of ligands at an air–water interface, we extended the model we previously used for studying the assembly of polymer-grafted NCs at polymer interfaces[15]. The long hydrophilic (H$_i$) and the short hydrophobic (H$_O$) ligands were treated as Kremer–Grest bead-chains[27] of lengths $l_{H_i}$ = 6 (unless otherwise stated) and $l_{H_o}$ = 3 beads. Each bead is of size $\sigma$ and represents a short segment of the ligand (Fig. 2a).

In our model, adjacent beads of the chains representing bonded segments interact with each other through finitely extensible non-linear elastic (FENE) springs and Weeks–Chandler–Anderson (WCA) potentials[28]. The FENE potential, which prevents bonded segments from stretching beyond a cutoff distance, is given by:

$$U_{FENE}(r, k, R_0) = -\frac{k}{2}R_0^2 \ln\left[1 - \left(\frac{r}{R_0}\right)^2\right] \quad (1)$$

where $r$ is the separation distance between the segments, $k = 30\varepsilon\sigma^{-2}$ is the spring constant, $R_0 = 1.5\sigma$ is the maximum possible length of the spring, and $\varepsilon$ is an energy parameter. The WCA potential, which models excluded-volume interactions between the bonded segments, can be conveniently presented in the form of a cut-and-shifted Lennard-Jones (LJ) potential, as given by

$$U_{LJ}(r, \sigma, \varepsilon, r_c) = \begin{cases} 4\varepsilon\left[\left(\frac{\sigma}{r}\right)^{12} - \left(\frac{\sigma}{r}\right)^6 - \left(\frac{\sigma}{r_c}\right)^{12} + \left(\frac{\sigma}{r_c}\right)^6\right] & r < r_c \\ 0 & r \geq r_c \end{cases} \quad (2)$$

where $r_c = 2^{1/6}\sigma$ is the cutoff distance of this short-range repulsive potential.

The liquid and vapor phases of water forming the gas–liquid interface were also treated using coarse-grained beads of size $\sigma$. The beads interact with each other via an attractive LJ potential $U_{LJ}(r, \sigma, \varepsilon, r_c = 2.5\sigma)$, which models both the attractive and the excluded-volume interactions as a result of the larger cutoff of $r_c = 2.5\sigma$. The simulations were carried out in the canonical (NVT) ensemble at a temperature of $0.7\,\varepsilon k_B^{-1}$ and a total number density of 0.4 beads $\sigma^{-3}$, which together with the above interaction potentials led to stable liquid–vapor phases (the number densities of liquid and vapor phases are about 0.79 beads $\sigma^{-3}$ and 0.01 beads $\sigma^{-3}$, respectively).

The nonbonded interactions between polymer segments, and between polymer segments and the surrounding fluid were treated based on their mutual miscibility. Chains of the same ligand type were considered fully miscible with each other. So, pairs of segments within a chain or across chains of the same type interacted with each other via the LJ potential $U_{LJ}(r, \sigma, \varepsilon, r_c = 2.5\sigma)$ containing attractive and repulsive components. Chains of different (specifically the hydrophilic and hydrophobic ligands) types were considered mutually immiscible, so their segments interacted with each other via the WCA potential $U_{LJ}(r, \sigma, \varepsilon, r_c = 2^{1/6}\sigma)$, which accounts for excluded-volume interactions only. The hydrophilic and hydrophobic ligands were assumed to be fully miscible and immiscible with the water phase, respectively. Thus, their interactions with the solvent beads were treated using the LJ potential $U_{LJ}(r, \sigma, \varepsilon, r_c = 2.5\sigma)$ and the WCA potential $U_{LJ}(r, \sigma, \varepsilon, r_c = 2^{1/6}\sigma)$, respectively.

The Ag NCs were modeled as a rigid simple-cubic 7 × 7 × 7 lattices of coarse-grained beads, each of size $\sigma$ (Fig. 2a). The ligand chains were attached to the surface beads of the NCs via the combined FENE–WCA bonding potential. We attempted to distribute the chains as uniformly as possible on each face of the NC. We explored overall grafting densities of $\Gamma$ = 0.25–1.0 chains $\sigma^{-2}$ (which resulted in chain conformations spanning across mushroom to brush regimes) and number percentage of hydrophobic ligands of 25–92% (Supplementary Fig. 3). The excluded volume interactions between the NCs and surrounding ligands and between NCs and water were treated using a

WCA potential $U_{LJ}(r, \sigma, \varepsilon, r_c = 2^{1/6}\sigma)$. The attractive interactions between NCs were calculated from the summation of the LJ potential $U_{LJ}(r, \sigma, \varepsilon, r_c = 2.5\sigma)$ acting between all pairs of beads across the interacting NCs. For more details, see refs. 5,7.

## Free energy calculation

To understand the orientational and assembly behavior of NCs, we performed two sets of free energy calculations. In both sets of calculations, the NCs were constrained at their "equilibrium" positions relative to the interfacial plane (i.e., normal distance $z$ from this plane), obtained by computing the ensemble averaged $\langle z \rangle$ of freely mobile NCs.

The first set of calculations involved carrying out MD simulations of a single NC for computing its orientational free energy as a function of Euler angles $(\theta, \varphi)$ (rotations about the internal axes parallel to $x$ and $y$ axes), specifically to determine the relative free energies of the idealized face-up, edge-up, and vertex-up orientations (Fig. 2d). To this end, we first prepared a well-equilibrated system of a single NC constrained in the pure face-up orientation $\left( \theta = 0^o, \varphi = 0^o \right)$ constrained at position $z = -1.9\sigma$ (i.e., $\langle z \rangle$ of the nanocube if it were freely mobile) relative to the air–water interface. Subsequently, we performed a very long MD simulation of the system in which the NC was first rotated around its center in steps of $\Delta\theta = 5^o$ (holding $\varphi = 0^o$) until it reached the pure edge-up orientation of $(45^o, 0^o)$ and then rotated about a different axis in steps of $\Delta\varphi = 5^o$ (holding $\theta = 45^o$) until the NC reached the pure vertex-up orientation of $(45^o, 45^o)$. In each rotation step, the NC was rotated with constant angular velocity over a period of 0.1 million timesteps and then held fixed for 0.6 million timesteps at each rotated angle. The ensemble-averaged torque component $\langle \mathcal{T}_\alpha(t) \rangle$ experienced by the entire NC (including grafts) in the direction of rotation $\alpha$ was measured from the last 0.5 million timesteps. The orientational free energy of NC (relative to its initial orientation) $\Delta F$ can then be obtained by simply integrating this torque component over the rotational angle as:

$$\Delta F(\alpha) = -\int_0^\alpha \langle \mathcal{T}_\alpha(\alpha) \rangle \mathrm{d}\alpha \qquad (3)$$

The second set of calculations involved MD simulations for computing the potential of mean force (PMF) of two NCs constrained in their preferred orientations (obtained from the orientational analysis of freely mobile NCs and confirmed by NC assembly) as a function of their closest surface–surface separation distance $d$ (Fig. 2e and Supplementary Fig. 14). These simulations were started from configurations generated during the equilibration step in which the two NCs were rotated to their preferred orientations and were placed in a face–face or edge–edge configuration with a separation distance of $d_0 = 10\sigma$. Keeping one NC fixed, the center of the other NC was brought closer to the fixed NC in equidistant steps of $\Delta d = 0.5\sigma$ until their closest surface–surface distance reached $d = 0$. During each displacement step, the center of the moving NC was moved at a constant speed for 0.1 million timesteps to its new location and then held fixed for 0.6 million timesteps at that location; the ensemble-averaged normal force $\langle f_d(t) \rangle$ experienced by the entire NC (including grafts) was measured from the last 0.5 million timesteps. At their initial position $d_0$, the two NCs were separated by a sufficiently large distance so that the net force $f_d(d_0)$ experienced by the moving NC is expected to be zero. The PMF can then be computed by simply integrating the normal force component over the traveled distance:

$$\Delta F(d) = -\int_{d_0}^d \langle f_d(d) \rangle \Delta d \qquad (4)$$

## Data availability

The data that support the findings of this study are available from the corresponding authors upon request. Source data are provided with this paper.

## Code availability

The code used in this study is available from the corresponding authors upon request.

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

## Acknowledgements

We thank the National Science Foundation, UCSD MRSEC DMR-2011924 for financial support. The authors acknowledge the use of facilities and instrumentation supported by the National Science Foundation through the UC San Diego Materials Research Science and Engineering Center (UCSD MRSEC) with Grant DMR-2011924 (A.R.T.) and the San Diego Nanotechnology Infrastructure (SDNI) of UCSD, a member of the National Nanotechnology Coordinated Infrastructure, which is supported by the Grant ECCS-2025752.

## Author contributions

Conceptualization: G.A., A.Frano., and A.R.T. Methodology: R.B., Y.W., Y.Z., Q.Y., G.A., A.Frano, and A.R.T. Investigation: R.B., D.L., Y.W., Y.X., Y.Z., A.D.Fuqua, W.S., and Z.Y. Visualization: R.B., Y.W., Y.Z., Q.Y., G.A., and A.R.T. Funding acquisition: G.A., A.Frano, and A.R.T. Project administration: G.A., A.Frano, and A.R.T. Supervision: G.A., A.Frano, and A.R.T. Writing (original draft): R.B., Y.W., Y.Z., G.A., and A.R.T. Writing (review and editing): R.B., Y.W., Y.Z., Q.Y., G.A., A.Frano, and A.R.T.

## Competing interests

The authors declare no competing interests.
