## [Peer Review File · Nature Communications]

Self-Assembly of Nanocrystal Checkerboards via Non-Specific InteractionsREVIEWER COMMENTS

Reviewer #1 (Remarks to the Author):

This paper discusses an interesting interfacial assembly approach where polymer-coated nanocubes spontaneously assemble into nonconventional checkerboard structures. The integration of computation, experiment, and machine learning based structural identification (of checkerboard motifs from electron microscopy images) is great. I am particularly intrigued by the mechanism: the polymer coatings that are isotropic (not through a more conventional regioselective manner) can adapt and become effectively segregated more towards the facets as the cubes approach each other in solution, guiding the directional assembly. I would recommend its publication given the following modifications.

1. Could the authors discuss more why the assembled checkerboard structures are significant for properties, and what sized assembly is needed to realize such properties? Only a few structural motifs would be sufficient or an extended periodic lattice structure is required to exhibit the desirable property? It would be also great if there are some property characterizations of the assembled structures.

2. In the experimental images, the percentage of the checkerboard motifs is not high (also related is the 3% phase space of success for checkerboard structure). Could the authors explain why the checkerboard structures are not large-scale or high yield? Due to kinetics or even thermodynamically the structure is not easily stable? For example, in Fig 2B, the phase diagram only shows the stable structures for each assembly condition. But are there other metastable structures that could be kinetically more favorable? If there are, what are the energy barrier or energy difference between the stable and metastable structures (i.e., are there interconversions between multiple structures in a condition)? Are there ways such as annealing to make the self-assembled structures of larger-size or higher purity?

3. In the simulation, was size, shape, or polymer coating heterogeneity considered? Would those factors affect the predicted stable structures? Was surface fluctuation (i.e., the fact that the liquid surface is not flat but undulated) of the liquid-air surface considered?

Reviewer #2 (Remarks to the Author):

Arya and Tao and their coworkers demonstrated that by combined experimental and simulation data presented in their study reveals that bespoke, precisely interconnected Ag NCs can be synthesized through non-specific chemical interactions. They showed that the control of interfacial, interparticle, and intermolecular forces is crucial in this approach to achieve the self-assembly of low to intermediate NC density mesophases. Utilizing CG models as a computational tool, they were able to rapidly assess the phase space and determine the essential parameters of NC building blocks such as chemical composition, nanoscale dimensions, and surface graft densities that are necessary for the formation of periodic structures.

The authors have produced a well-constructed and well-organized manuscript. The combination of systematic study and both experimental and simulation techniques have enabled the elucidation of the mechanism for controlling self-assembly through the utilization of two ligands, thiolated PEG and alkanethiol. The authors had previously published research in 2012 in Nature Nanotechnology where they introduced the concept of controlling the self-assembly of Ag NCs through the use of PVP and demonstrated edge-to-edge assembly. The present investigation expands upon prior research, but the utilization of two ligands in the current study and the refined control of self-assembly does not result in significant novel observations that merit publication in the journal Nature Communications. In addition, while the SEM images are visually appealing, they do not meet the standards of Nature Communications as they only depict small regions with checkerboard structures and exhibit imperfect conditions. Ultimately, the manuscript, as it stands, may not adhere to the criteria set forth by Nature Communications unless the authors can exhibit the capability to self-assemble checkerboard structures across a significantly expansive area with minimal flaws. This demonstration will

undoubtedly justify the merit of publication in the prestigious journal of Nature Communications.

Reviewer #3 (Remarks to the Author):

the use of two-graft Ag nanocubes to tune interparticle interactions and achieve a checkerboard lattice. The authors used CG MD simulations to build a structural phase diagram of different nanocube orientation geometries and assembly configurations based on different % ratios of hydrophilic/hydrophobic ligands and the overall graft density. This enabled them to identify the experimental conditions necessary to access the checkerboard lattice, which required a highly specific and narrow assembly conditions. The authors have proposed an interesting concept which demonstrates potential to access various inorganic nanocrystal lattices and will be of interest to a broad audience. The experimental observations are also well-supported with extensive MD simulations and calculations. As such, the manuscript can be considered for publication upon major revision, with several questions that require further elaboration.

- 1) In the discussion for Figure 1, can the authors elaborate on what they mean by 'iterative feedback between CG MD simulations and NC assembly experiments was used to converge upon the appropriate assembly conditions'? The use of 'iterative feedback' implies some sort of repeating feedback loop and it would be good if the authors can clarify how this process is performed.
- 2) Can the authors comment on the feasibility of large-area checkerboard lattice formation? It appears from Figure 1B that the checkerboard lattice self-assembly is still rather disordered and non-periodic.
- 3) How do the authors control or determine how the ligands are distributed on the nanocube surface e.g., whether they exist as Janus monolayers or are highly disordered or form local domains? In addition, do the authors have any experimental evidence on how the ligands are arranged on the nanocube surface?
- 4) On a similar note, how reproducible is the lattice formation using the two-graft nanocubes? Do the authors have experimental data of different batches of two-graft nanocubes?
- 5) The SEM image in Figure 1B has poor resolution. Can the authors replace with sharper images?
- 6) For Figure 2B, it is hard to interpret the difference in nanocube orientation between the ones denoted as squares, with and without the line in the middle.
- 7) Figure 3A-v appears more like a 1D face-face nanocube assembly than 2D face-face assembly.
- 8) The phrase 'Once formed, they and disturb the structural' found in the discussion for Figure 4 is a typo.

We would like to thank the editor and the reviewers for reading through our manuscript and providing their valuable opinions and suggestions. We have implemented the reviewers' suggestions in our revised manuscript and believe that the overall quality of the manuscript has improved significantly. Sections of the manuscript that have been modified are highlighted in the revised manuscript so that the editor and the reviewers can quickly identify these changes. We have also included a point-by-point response (in blue) to the reviewers' comments (original comments are in italics) to justify and further explain the changes made in the revised manuscript.

Reviewer #1:

1. *Could the authors discuss more why the assembled checkerboard structures are significant for properties, and what sized assembly is needed to realize such properties? Only a few structural motifs would be sufficient or an extended periodic lattice structure is required to exhibit the desirable property? It would be also great if there are some property characterizations of the assembled structures.*

We identified two key features of the checkerboard lattice that can be realized in a functional material. The first is that the Ag nanocube checkerboards support strong coupling of quadrupolar plasmonic resonances that give rise to unique optical behavior. Finite-difference time-domain simulations for an infinite 2D lattice of Ag nanocubes (**Fig. R1**) show a strong absorption peak corresponding to a coupled quadrupolar lattice resonance at 496 nm for an inter-nanocube gap spacing of 5.5 nm. The strong absorption peak is polarization independent due to the symmetry of the lattice, unlike the equivalent edge-to-edge type assembly for a 1-D chain of nanocubes which behave like a highly polarized optical waveguide. Our simulations also indicate that this coupling behavior requires interconnectivity between at least 5 nanocubes to achieve this particular quadrupolar lattice mode (**Fig. R2**). Thus, as the reviewer aptly points out, only a few structural motifs are needed for exploiting this optical behavior. However, these electrodynamic simulation results go beyond the scope of the manuscript and have not been included in the main text.

A second key feature is the porosity of the checkerboard lattice. The shaped pores generated by these structures could be used in biomolecular separations and capture. However, such an application likely requires an extended lattice to yield a robust membrane-like material.

Figure R1. (a) Model of the checkerboard pattern with the incident light direction indicated. (b) Zoom-in of the red box from figure 5a showing how gap size between the nanocubes are defined. (c) Absorption spectra of checkerboard pattern with different polarizations of incident light compared.

Figure R2. Absorption spectra of increasing number of nanocubes connected edge-to-edge illuminated from the z direction.

The following discussion has been added to the revised manuscript on Pg. 18: “The checkerboard-type structures achieved here have two key functional features: (i) shaped pores that have the potential to be used in chemical separations and/or capture, and (ii) strong coupling of quadrupolar plasmonic resonances that give rise to unique optical behavior (Fig. S18).”

2. *In the experimental images, the percentage of the checkerboard motifs is not high (also related is the 3% phase space of success for checkerboard structure). Could the authors explain why the checkerboard structures are not large-scale or high yield? Due to kinetics or even thermodynamically the structure is not easily stable? For example, in Fig 2B, the phase diagram only shows the stable structures for each assembly condition. But are there other metastable structures that could be kinetically more favorable? If there are, what are the energy barrier or energy difference between the stable and metastable structures (i.e., are there interconversions between multiple structures in a condition)? Are there ways such as annealing to make the self-assembled structures of larger-size or higher purity?*

While we agree that the domain sizes of the checkerboard motifs observed in our experiments are small, these motifs make up the majority of the assembled structures found in our best samples (see also our response to reviewer 2). However, we also observed that many of our experiments lead to 1D edge-edge and 1D face-face mesophases in coexistence with the checkerboard motifs. To

understand this, we carried out additional simulations with larger 16-nanocube systems to refine our mesophase diagram; note that in the original manuscript, a 4-nanocube system was used for constructing the phase diagram (Fig. 2B). We also examined much larger systems composed of 64 and 144 nanocubes, but due to their high computational cost, these simulations were much shorter in length and could not capture full assembly. Our newly reconstructed self-assembly phase diagram using 16 nanocubes is provided in **Fig. R4** (and also Fig. 4B)

Figure R4. Phase diagram constructed with 16 nanocubes.

We discovered that this new phase diagram has a transition band where multiple assembly morphologies coexist. Most notably, we find that for nanocubes grafted with $\sim 70:30$ hydrophobic to hydrophilic ligand ratio and ~ 0.75 normalized grafting density, the checkerboard phase coexists with the 1D edge-edge and 1D face-face phases, in remarkable agreement with our experimental findings. The following discussion has been added to the revised manuscript on Pg. 11:

Low magnification SEM images show that the checkerboard lattice formation is not limited to a small region of the sample, and rather, represent the major product of assembly (Fig. 4A). Although the domain sizes of the checkerboard motifs observed in the experiments were small, such assemblies can be captured from virtually everywhere on the macroscopic air-water interface (Fig. S10). We did observe, however, that many of our experiments led to 1D edge-edge and 1D face-face mesophases in coexistence with the checkerboard motifs. This is also consistent with our phase diagram, which shows a transition band where multiple assembly morphologies coexist. Most notably, we found that this coexistence is prominent for nanocubes grafted with a 7:3 hydrophobic to hydrophilic ligand ratio and ≈ 0.75 normalized graft density (Fig. 4B), confirming the sensitive nature of checkerboard lattice formation. While atomic force microscope characterization (Fig. S11) confirms that PEG chains are well distributed on the Ag nanocube surface, small variations in graft density or graft distribution (Fig. S12) increases the likelihood that 1D assemblies are formed.

Figure R5. Free energy of bringing two nanocube dimers into the checkerboard and triangular arrangements as a function of the edge-to-edge distance between two dimers.

We also believe that the sizes of the checkerboard domains are limited by defects, primarily vacancies (where a NC is missing from the lattice) and triangular arrangements (where three NCs assemble via edge-edge connections), as mentioned in Fig. 4 in the original manuscript. We believe such defects prevent NC attachment at the open edge sites of an existing nanocube cluster. To investigate the thermodynamics of triangular defect formation, we performed additional potential of mean force (free energy) calculations using a pair of nanocube dimers to compare the free energy of formation of a checkerboard motif against formation of triangular motif and other motifs like straight and zigzag lines (**Fig. R5**). We find that triangular motif has a similar free energy of formation ($-97 \pm 7 k_B T$) compared to the checkerboard cluster ($-108 \pm 11 k_B T$). This suggests that the formation of triangular motifs is in fact almost as favorable as the formation of the checkerboard motif and that there is an energy barrier to overcome for rearranging a triangular cluster into a checkerboard cluster. The straight and zigzag lines lead to smaller but still favorable free energies, suggesting that those motifs could also appear in nanocube assemblies.

This is consistent with our experiments on thermal annealing to remove these defects from the nanocube assemblies. We carried out thermal annealing of the Langmuir–Blodgett (LB) films under mild conditions (50 °C, 4 hours); longer thermal annealing times or increased temperatures leads to a shape change, where the nanocubes become rounded and approach a spherical shape. We observe that mild thermal annealing accelerates nanocube assembly, which normally takes up to 72 hours at room temperature. However, thermal annealing does not promote the assembly of larger scale, extended checkerboard lattices, as shown in **Fig. R6**. We also tried agitation annealing by sonicating the assembled checkerboard film to disassemble NCs and assembly defects. However, the redistributed NCs reassemble into checkerboard lattices after 24 hours without a significant difference in the quality or size of the checkerboard domains (**Fig. R7**). This data has been added to the supporting information in Fig. S19 and Fig. S20 in the revised manuscript.

Figure R6. Thermal annealing of the Langmuir–Blodgett (LB) films under mild conditions (50 °C, 4 hours).

Figure R7. Agitation annealing by sonicating the assembled checkerboard film to disassemble NCs and reassembly.

3. *In the simulation, was size, shape, or polymer coating heterogeneity considered? Would those factors affect the predicted stable structures? Was surface fluctuation (i.e., the fact that the liquid surface is not flat but undulated) of the liquid-air surface considered?*

In this study, only homogeneously sized and shaped nanocubes were considered in our simulations. We did however study two variation of uniform grafting patterns (Fig. S3 originally, now Fig. S5) in the original manuscript for the condition where checkerboard lattice was observed (normalized grafting density of 1 and number percentage of hydrophobic ligands of 75 %) and found that both led to checkerboards. To investigate, how heterogeneities in polymer grafting affect the assembly outcome, we examined eight additional grafting patterns for this checkerboard condition, where the grafts are more unequally distributed on the facets of the NC (**Fig. R8**). We observed that most of the grafting patterns did *not* lead to checkerboard assembly; only Pattern 8, where the PEG chains are in the middle of the facet, still led to checkerboard formation. Such a configuration of PEG ligands likely leads to a similar mechanism for NC interaction as in the original manuscript, where steric repulsion from PEG chains deters face-face NC interactions.

The surface fluctuations of the liquid-air surface are already considered in our simulations because the water molecules are modeled explicitly as Lennard-Jones particles (Fig. 2A). Therefore, the water-air surface is free to fluctuate and is indeed found to undulate over the simulation period.

Figure R8. Different grafting patterns for the condition of normalized grafting density = 1 and number percentage of hydrophobic ligands = 75%, where checkerboard assembly was observed for the two grafting patterns in Fig. S3 in the original manuscript.

In addition to heterogeneity in grafting patterns discussed above, we also showed in the original manuscript that variations in the length of the PEG chains, the net grafting densities, and the C_{16} :PEG ratios could also affect the propensity of checkerboard formation. In particular, we carried out simulations with four different PEG chain lengths (6-, 7-, 8-, and 9-bead long) and found that the system with 8-bead PEG ligands formed perfect checkerboard structures without defects (Fig. S13). We also showed that the checkerboard lattice requires nanocube surface ligation with 75% C_{16} and 25% PEG. Based on our simulations, an 8% decrease in the hydrophobic ligand fraction or a 25% decrease in the total ligand density leads to the formation of 1D edge-edge structures, while an 8% increase in hydrophobic ligand fraction or a 25% increase of the total ligand density leads to the formation of 1D face-face structures. In our experiments, however, graft density is largely determined by the radius of the polymer brush and radius of curvature of the nanocubes. It is difficult to precisely control surface grafting density due to inhomogeneities in the colloidal sample (distribution sizes/shapes) and in the polymer graft (distribution of molecular weights). All these results from simulations suggest that the heterogeneity of polymer grafts on the NC surface likely contribute to the formation of small checkerboard domains and defect formation.

Reviewer #2:

The present investigation expands upon prior research, but the utilization of two ligands in the current study and the refined control of self-assembly does not result in significant novel observations that merit publication in the journal Nature Communications. In addition, while the SEM images are visually appealing, they do not meet the standards of Nature Communications as they only depict small regions with checkerboard structures and exhibit imperfect conditions. Ultimately, the manuscript, as it stands, may not adhere to the criteria set forth by Nature Communications unless the authors can exhibit the capability to self-assemble checkerboard structures across a significantly expansive area with minimal flaws.

We are assuming that the reviewer is referring to our 2012 *Nature Nano* paper, which is co-authored by a subset of the current authors. This work, where we demonstrated tunable self-assembly of face-face and edge-edge strings of single-ligand grafted nanocubes by simply modulating the length of the grafted polymers, is part of the broad, long-term vision for controlling the self-assembly of grafted nanoparticles.

In the current manuscript, we have taken a large step towards this vision by demonstrating a computation-guided process that combines theoretical predictions of assembly, identifiable knobs to control the assembly, and analysis methods that quantify the assembly. Moreover, we generate a highly architected structure that has never been observed in colloidal assemblies. The referee ignores this accomplishment. The interplay between polymers, organic ligands, and nanoparticles is at the very forefront of research on mesoscale soft/hard matter hybrids, and there is still a fundamental lack of understanding regarding how to program assembly in these materials systems.

Moreover, the creation of a large-scale checkerboard lattice (which does not even exist in solids) does not preclude such structure from being useful, from a materials standpoint. As demonstrated in our response to Reviewer #1 Q1, the optical properties of even a 3x3 domain of the checkerboard lattice should give rise to quadrupolar plasmonic coupling.

Finally, we wish to point out that while it is true that the checkerboard lattices only form in small domains (e.g., 3x3 and 4x4 arrays), the checkerboards certainly do not only form over small regions of the air-water interface. Low magnification SEM images show that the checkerboard lattice formation is not limited to a small region of the sample, and rather, represent the major product of assembly. (**Fig. R9**) Such assemblies can be captured from virtually everywhere on the air-water interface of the Langmuir–Blodgett trough (which is 36.5 cm x 7.5 cm in area), as seen in the **Fig. R10**. These data have been added to Fig. 4A and the supporting information in S10.

Figure R9. Low magnification SEM image of checkerboard mesophase. The bottom image corresponds to a magnified image of the region outlined in yellow in the top image.

Figure R10. Langmuir-Blodgett trough with checkerboard mesophase.

Reviewer #3:

- 1) *In the discussion for Figure 1, can the authors elaborate on what they mean by ‘iterative feedback between CG MD simulations and NC assembly experiments was used to converge upon the appropriate assembly conditions’? The use of ‘iterative feedback’ implies some sort of repeating feedback loop and it would be good if the authors can clarify how this process is performed.*

The phrase “iterative feedback” refers generically to the feedback loop below:

where experimentally determined parameters (e.g., polymer graft length, chemistry, and density) are used to implement computational models and, in turn, simulation results are used to provide design feedback to the experiments. In the specific context of generating checkerboard lattices, we started by synthesizing nanocube building blocks within only a small phase space (e.g., number percentage of hydrophobic ligands = 75 %). We next used coarse-grained MD simulations to model nanocube self-assembly over a larger phase space to identify the critical influence of parameters such as ligand density and ligand percentage. We then carried out assembly experiments to validate the simulation results. If the experimental results agree with the simulations, we then use simulations to identify the experimental parameters required for checkerboard assembly. This process was repeated until we were able to develop a protocol identifying the experimental parameters (e.g., ligand exchange solvent and feedstock concentrations) for checkerboard assembly.

- 2) *Can the authors comment on the feasibility of large-area checkerboard lattice formation? It appears from Figure 1B that the checkerboard lattice self-assembly is still rather disordered and non-periodic.*

There are two main reasons why we are not able to achieve checkerboard lattices with large domain sizes: i) the nanocubes may possess heterogeneous surface grafting densities, and ii) multiple metastable mesophases coexist with the checkerboard phase. These reasons are discussed in more detail in our response to questions 2 & 3 from Reviewer #1 and our response to Reviewer #2. To clarify this in our manuscript, we have significantly revised Figure 4 and the corresponding discussion. The following paragraph has been added to Pg. 11 of the revised manuscript:

Low magnification SEM images show that the checkerboard lattice formation is not limited to a small region of the sample, and rather, represent the major product of assembly (Fig.4A). Although the domain sizes of the checkerboard motifs observed in the experiments were small, such assemblies can be captured from virtually everywhere on the macroscopic air-water interface (Fig. S16). We did observe, however, that many of our experiments led to 1D edge-edge and 1D face-face mesophases

in coexistence with the checkerboard motifs. To understand this, we carried out additional simulations with larger 16-nanocube systems to refine our mesophase diagram (Fig.4B). We discovered that this highly resolved phase diagram has a transition band where multiple assembly morphologies coexist. Most notably, we find that this coexistence is prominent for nanocubes grafted with a 7:3 hydrophobic to hydrophilic ligand ratio and ≈ 0.75 normalized graft density, confirming the sensitive nature of checkerboard lattice formation. While atomic force microscope characterization (Fig. S18) confirms that PEG chains are well distributed on the Ag nanocube surface, small variations in graft density or graft distribution (Fig. S17) increases the likelihood that 1D assemblies are formed.

3) *How do the authors control or determine how the ligands are distributed on the nanocube surface e.g., whether they exist as Janus monolayers or are highly disordered or form local domains? In addition, do the authors have any experimental evidence on how the ligands are arranged on the nanocube surface?*

We believe that the two ligands are uniformly distributed on the facets of the functionalized NC surface and that these ligands do not form Janus-type monolayers. To demonstrate this indirectly, we assembled the NCs through gravitational sedimentation and evaporative assembly (Fig. S9):

Figure R11. SEM images of AgNCs functionalized with 50 μ M PEG20k and 6 μ M C16 in feedstock assembled by gravitational sedimentation and dropcast on Si substrate (scale bar 500 nm).

As seen in the SEM images above, the NCs assembled into high-density, closed-packed 3D superstructures, which are different as Janus-type structures, where the nanocubes are selectively functionalized on only a few facets typically results in 1D (2 facets) or 2D (4 facets) superstructures¹, and are indicative of surface to be homogeneous.

¹ (1) Rycenga, M.; McLellan, J. M.; Xia, Y. Controlling the Assembly of Silver Nanocubes through Selective Functionalization of Their Faces. *Advanced Materials* 2008, 20 (12), 2416-2420. DOI: <https://doi.org/10.1002/adma.200800360> (accessed 2023/08/03).

Figure R12. AFM phase contrast image of edge-edge aligned Ag nanocubes.

Additional simulations carried out in our manuscript revisions also show that the checkerboard lattice is only achievable with i) homogeneously distributed ligands, or ii) patchy nanocubes where the PEG ligands are attached closed to the center of each facet. (See response to Reviewer #1, Q2) However, AFM phase contrast imaging indicates that there is no clustering of PEG ligands in the center of each facet and the ligand shell is homogeneous (Fig. R12). This has been added to the supporting information in S18. Together, these results suggest that the two ligands indeed uniformly distributed over the Ag nanocube surface.

- 4) *On a similar note, how reproducible is the lattice formation using the two-graft nanocubes? Do the authors have experimental data of different batches of two-graft nanocubes?*

The versatility and reproducibility of our approach is proved through using NCs of different batches composed of different NC sizes (Fig. S5) and PEG grafts of varying Mw (10-54K g/mol, Fig. S7 and S8). Fig. S8 also shows different batches of NCs (B1-B8) grafted with 50 μ M:6 μ M (PEG20k:C16) that all assembled into checkerboard mesophase. While reproducible, it should be noted that the checkerboards are commonly observed alongside 1D edge-edge assemblies, which is a metastable mesophase that coexists with the checkerboard phase.

- 5) *The SEM image in Figure 1B has poor resolution. Can the authors replace with sharper images?*

Fig. 1B was updated with a SEM image of better resolution.

- 6) *For Figure 2B, it is hard to interpret the difference in nanocube orientation between the ones denoted as squares, with and without the line in the middle.*

In Fig. 2B, the orientations of the nanocubes indicated by the gray geometries are shown as top view of the nanocubes at the interface (i.e., as seen looking from the air, down to the liquid). Therefore,

the orientation denoted as squares without a line in the middle represents face-up, while the one with a line in the middle represents the edge-up orientation.

7) *Figure 3A-v appears more like a 1D face-face nanocube assembly than 2D face-face assembly.*

As the reviewer pointed out, the SEM image in Fig. 3A-v appears as a 1D face-face phase by human eyes, however, it was identified to be 2D face-face quantitatively based on the angular cross-correlation (ACC) analysis on the two-dimensional Fourier transform (FT) of the SEM image.

8) *The phrase 'Once formed, they and disturb the structural' found in the discussion for Figure 4 is a typo.*

We have corrected this sentence to "Once formed, they disrupt the structure of..." in the discussion for Figure 4 in the revised manuscript.

REVIEWER COMMENTS

Reviewer #1 (Remarks to the Author):

The authors have tried their best to address my comments. While I agree with the other two reviewers that the paper's weakness is that the checkerboard structure is of low yield, I feel the theoretical picture of multiphysics process is still inspirational enough to be published on Nature Communications.

Reviewer #2 (Remarks to the Author):

The questions have been accurately addressed, but I recommend reconsidering the title, as the structures appear to be a haphazard assembly of edges rather than organized in a uniform checkerboard pattern.

Reviewer #3 (Remarks to the Author):

I have carefully examined the responses provided by the authors to all three reviewers. I must admit that I remain unconvinced regarding the novelty of this work. While it is true that domains of checkerboard lattices were indeed observed in some areas of the samples, the low magnification SEM images suggest that these checkerboard lattices continue to exhibit a significant level of disorder and lack the desired periodicity. The authors primarily base their discussion on the observations made within these limited areas of small domains of checkerboard lattices. Moreover, the authors have not adequately furnished additional demonstrations and discussion on the resulting properties of the different types of checkerboard lattices. In particular, the absence of comprehensive optical characterization at the ensembled levels has not provided convincing evidence of the distinctive optical behavior and practical utility of these lattices.

Point-by-point response – January 3, 2024

Reviewer #3

While it is true that domains of checkerboard lattices were indeed observed in some areas of the samples, the low magnification SEM images suggest that these checkerboard lattices continue to exhibit a significant level of disorder and lack the desired periodicity. The authors primarily base their discussion on the observations made within these limited areas of small domains of checkerboard lattices.

While we agree that the domain sizes of the checkerboard motifs observed in our experiments are small (4x4), these motifs make up the majority of the assembled structures collected at the air-water interface. These observations are also consistent with coarse-grained molecular dynamics (CG-MD) simulations, which are also carried out for a 16-nanocube system. We firmly believe that the self-assembly mechanism presented in our manuscript is accurate and that our computation-guided approach presents a novel strategy for interfacial assembly, despite a lack of long-range periodicity due to the reasons outlined in our previous response. In addition, our optical measurements indicate (as discussed below) that the functional optical properties of the Ag nanocube (AgNC) checkerboards are exhibited over macroscopic length scales.

Moreover, the authors have not adequately furnished additional demonstrations and discussion on the resulting properties of the different types of checkerboard lattices. In particular, the absence of comprehensive optical characterization at the ensembled levels has not provided convincing evidence of the distinctive optical behavior and practical utility of these lattices.

In the second revision of our manuscript, we have included additional optical measurements and electrodynamic simulations to demonstrate the distinct optical behavior of the edge-edge and checkerboard nanocrystal lattice. In our first revision, we identified electromagnetic coupling as a key functional feature of the checkerboard lattice. The AgNC checkerboards support strong coupling of quadrupolar plasmon resonances that give rise to unique optical behavior associated with higher-order lattice resonances, namely low radiative losses that stem from diffractive-type coupling (in comparison to dipole-dipole coupling). This type of quadrupolar lattice resonance is of particular interest for nanolasing and optical nanocavities because they give rise to high-Q optical resonances. In addition, quadrupolar lattices can also give rise to novel light-matter interactions such as the excitation of “dark” plasmonic modes and the creation of topological plasmonic modes.

In this current revision, we have included additional experiments in the Supporting Information to provide evidence of these quadrupolar lattice resonances (Figure S5, also included at the end of this response). Namely, we carried out *in situ* optical reflection measurements of AgNCs that are in two configurations: i) isotropically distributed and well-separated (> 100 nm) AgNCs, and ii) edge-edge/checkerboard assemblies. To validate these measurements, we first built an appropriate electrodynamic model to carry out finite-difference time-domain (FDTD) simulations of the AgNCs. In our previous manuscript revision, these studies were carried out using a model with a homogeneous dielectric environment (air) for the AgNCs. To match the conditions for *in situ* interfacial measurements, a heterogeneous dielectric environment representing the air-water interface is necessary. We first carried out a series of simulations to determine where to place a

planar air-water interface relative to the height of the floating AgNC. Optical reflection spectra were simulated for isolated AgNCs with varying heights, h , of the air-water interface along the z -direction of the model. Simulation results, where AgNCs are modelled with a spacing of >100 nm to approximate little-to-no plasmonic coupling, shows a dip in the reflection spectra that redshifts from $\lambda=410$ nm to $\lambda=459$ nm as the height of the air-water interface increases from the bottom facet of the AgNC ($h=0$ nm) to the top facet of the AgNC ($h=80$ nm), respectively. This is attributed to peak splitting of the quadrupolar localized surface plasmon resonance (LSPR) mode of an isolated AgNC due to anisotropic dielectric environment generated by partial submersion of the nanocube in water.

Next, we obtained *in situ* reflection UV-Vis-NIR measurements for floating AgNCs on the air-water interface. Reflection spectra were obtained by illuminating the floating AgNC layer using a tungsten-halogen lamp (~ 10 mW, ThorLabs). The reflected signal was captured through a fiber optic reflection probe and measured by CCD spectrometer (ThorLabs). The cross-sectional area of the measured film area was measured at 8.6 mm², covering a macroscopic area (i.e. not focused on one or a few small checkerboard domains). Reflectance spectra were obtained by averaging 100 acquisitions with an acquisition time of 20-40 ms as needed to maximize signal without oversaturating the detector. Dispersed AgNC films were experimentally obtained using as-made AgNCs that are capped with PVP. The measured reflection spectra of these AgNCs when dispersed on the air-water interface show a reflection dip at $\lambda = 442$ nm, consistent with the simulation results for $h = 70$ ($\lambda = 451$ nm), indicating that 89% of the AgNC volume is submerged below the air-water interface. This is also consistent with CG-MD simulation results in Figure 2 of the main manuscript. This air-water interface height of $h = 70$ was then used to simulate the reflectance of checkerboard and edge-edge AgNCs with a 3.5 nm interparticle gap.

We then carried out FDTD simulations and *in situ* reflection UV-Vis-NIR measurements for the checkerboard lattices. The simulated spectrum shows a reflection dip at $\lambda = 660$ nm — well beyond what is observed for dispersed AgNCs — that is attributed to quadrupole-quadrupole LSPR coupling. This is consistent with a large absorption peak also observed in FDTD simulations. The simulated reflection spectrum also exhibits a reflection peak shoulder at $\lambda = 516$ nm which is attributed to weaker plasmonic coupling between higher order modes. Both features match well with the experimental data obtained for edge-edge and checkerboard assembled AgNCs. These AgNCs films were obtained with the same surface chemistries described in the main manuscript (50 μ M PEG20k-SH and 6 μ M C₁₆-SH). In comparison to the electrodynamic model, the experimentally obtained films adopt much lower AgNC film densities and exhibit variances in edge-edge AgNC connection geometries, resulting in much broader spectral features. However, the observation of a reflection dip at $\lambda = 660$ nm strongly indicates that the optical properties of the macroscopic AgNC assembly is dominated by quadrupole-quadrupole plasmonic coupling that results directly from edge-edge interactions in a periodic lattice.

These data and corresponding discussion have been added to the Supporting Information in sections S1.10, S2.4, and S5.

Figure S5: FDTD simulations and UV-Vis-NIR reflection measurements of interfacial AgNC assemblies. (A) Diagram of FDTD model showing the varying height, h , of the air-water interface with the z -direction. (B) Simulated reflection spectra of a dispersed AgNC film with varying air-water interface locations as indicated by h . (C) Experimental (red line) and simulated (black line, $h=70$ nm) reflection spectra of dispersed well-separated AgNCs at the air-water interface. (D) Experimental (red line) and simulated (black line, 3.5 nm spacing) reflection spectra for edge-edge assembled AgNC at the air-water interface. (E) Corresponding SEM images of the dispersed AgNC film from panel C after transfer onto a solid Si substrate. (scale bar = 20 μm , inset scale bar = 1 μm). (F) Corresponding SEM images of the edge-edge assembled AgNCs measured in panel D after transfer onto a solid Si substrate. (scale bar 20 μm , inset 1 μm).

REVIEWERS' COMMENTS

Reviewer #3 (Remarks to the Author):

Dear Editor,

The authors have sufficiently addressed my questions. I have no other questions.